# Effect of crown stiffness and prosthetic screw absence on the stress distribution in implant-supported restoration: A 3D finite element analysis

Ettore Epifania[1], Alessandro E. di Lauro[1], Pietro Ausiello[1]*, Alessia Mancone[1], Franklin Garcia-Godoy[2], João Paulo Mendes Tribst[3]

1 Department of Neurosciences, Reproductive and Odontostomatological Sciences, School of Dentistry, University of Naples Federico II, Naples, Italy, 2 Department of Bioscience Research, College of Dentistry-University of Tennessee Health Science Center, Memphis, TN, United States of America, 3 Department of Reconstructive Oral Care, Academic Centre for Dentistry Amsterdam (ACTA), Universiteit van Amsterdam en Vrije Universiteit Amsterdam, Amsterdam, The Netherlands

* Pietro.ausiello@unina.it

## Abstract

This in-silico investigation evaluated the mechanical impact of Morse tape implant-abutment interface and retention system (with and without screw) and restorative materials (composite block and monolithic zirconia) by means of a three-dimensional finite element analysis (3D-FEA). Four 3D models were designed for the lower first molar. A dental implant (4.5 × 10 mm B&B Dental Implant Company) was digitized (micro CT) and exported to computer-aided design (CAD) software. Non-uniform rational B-spline surfaces were reconstructed, generating a 3D volumetric model. Four different models were generated with the same Morse-type connection, but with a different locking system (with and without active screw) and a different crown material made of composite block and zirconia. The D2 bone type, which contains cortical and trabecular tissues, was designed using data from the database. The implants were juxtaposed inside the model after Boolean subtraction. Implant placement depth was simulated for the implant model precisely at crestal bone level. Each acquired model was then imported into the finite element analysis (FEA) software as STEP files. The Von Mises equivalent strains were calculated for the peri-implant bone and the Von Mises stress for the prosthetic structures. The highest strain values in bone tissue occurred in the peri-implant bone interface and were comparable in the four implant models (8.2918e-004–8.6622e-004 mm/mm). The stress peak in the zirconia crown (64.4 MPa) was higher than in the composite crown (52.2 MPa) regardless of the presence of the prosthetic screw. The abutment showed the lowest stress peaks (99.71–92.28 MPa) when the screw was present (126.63–114.25 MPa). Based on this linear analysis, it is suggested that the absence of prosthetic screw increases the stress inside the abutment and implant, without effect on the crown and around the bone tissue. Stiffer crowns concentrate more stress on its structure, reducing the amount of stress on the abutment.

**Data Availability Statement:** All relevant data are within the manuscript.

**Funding:** The author(s) received no specific funding for this work.

**Competing interests:** The authors have declared that no competing interests exist.

## Introduction

Surgical application of dental titanium implants is nowadays considered a predictable therapy to rehabilitate partial and full edentulous patients [1]. The major purpose of this long-term therapy is to rehabilitate the chewing function and biologically maintain a sound bone level around the fixtures. In fact, marginal bone loss is a percentage directly related to implant failure [2]. However, the etiology of bone loss is multifactorial. According to Albrektsson et al. [3], in the first year of function, the clinically acceptable threshold of bone loss around a dental implant is 1.5 mm, with approximately every year of bone loss of 0.2 mm without infection.

The biomechanical behavior of the bone around a single dental implant restored with a crown depends on different factors [4]. One of these is the final rigidity of the implant-supported crown system. It has been shown that close contact between the dental implant and bone tissue transfers occlusal vectorial load effects directly to the bone itself. In many conditions, this chewing load may exceed the physiological elasticity of bone, accelerating bone resorption also because of a missing natural shock absorber like the periodontal ligament [5]. Another factor that biomechanically contributes to the stress concentration in bone tissue is the cervical level and the angle of the lateral surface of the prosthetic connection. Several authors [6] investigated this topic through numerical simulation and they found that implants with 10˚ and 20˚ neck designs should be chosen as an alternative of straight platforms to better redistribute stress. In this sense, other points need to be strictly investigated: the role of the implant-abutment connection, external or internal, and of the loosening of the prosthetic screw [7] on the microstrain distribution and its effect on the bone tissue.

During the rehabilitation of missing teeth, the placement of implant-supported restorations became a standard clinical practice, and different systems and connections are available to be used with shared clinical indications [8]. The morse-tape design containing a screw-retained restoration is encouraged due to more advantageous stress distributions and reduced risk for leakage in comparison with external and internal hexagonal connections [9, 10]. Although morse-taper prosthetic connection offers a steady locking mechanism that can decrease micro-motion and micro gaps, stresses concentrated at the screw are still noticeable [10]. However, the morse-taper system has a greater torque maintenance capacity and is associated with a smaller component conformation by excessive occlusal forces [8]. Aiming to overcome any limitation caused by the presence of prosthetic screw, some implant systems are available without this structure, using solid and friction retained abutment designs [11]. The disadvantage of this condition is less reversibility for implant-supported restorations, while the mechanical advantage is still not clear. To investigate some mechanical aspects related to these problems, a new approach has been proposed using an in-silico investigation.

The three-dimensional (3D) finite element analysis (FEA) is a theoretical numerical analysis that is useful to investigate stresses and strains of complex systems. It is properly applied also in biomedicine and in different fields of dentistry [12–19] to study the internal and marginal adaptation of materials and dental tissues. Furthermore, the literature widely employs three-dimensional finite element analysis (FEA) approach, for a more accurate simulation of the stress distribution within the implant system compared to traditional analytical methods [10, 19]. It was reported that FEA results can contribute to a better understanding of the biomechanical behavior of dental implant, frameworks and different rehabilitation designs which can inform the development of more effective and durable implant designs [19, 20].

In this study, it was considered the effects of occlusal loading on strain and stress development in a morse-type implant-abutment connection where taper surfaces are connected by a cold-welding effect and the two parts engaged together, dependently on different crown

material stiffness (E). The null hypotheses were: 1) screw effect inside the connection is negligible; 2) crown restoring material does not influence the implant-bone interface.

## Methods

A three-dimensional implant, abutment and screw shapes (Dura-vit 3P B&B DENTAL, Bologna, Italy) were digitized using technique of reverse engineering (Micro-CT, Skyscan 1172) according to the manufacturer's dimensions. For that, point clouds have been exported to the Geomagic Studio® dashboard, where 3D STL network was generated. Feature recognition algorithms from Geomagic Studio® software, were then applied to reduce sharp boundaries and cross-segmented curves. The STL was then exported to Computer Aided Design (CAD) software (Rhinoceros version 5.0 SR8, McNeel, Seattle, USA) and the plugin 'ReduceMesh' was used with 45% significance, smoothing the assembly with the total normal faces directed in the same path. The NURBS (non-uniform rational B-spline) shells were recreated from the STL producing a 3D volumetric model analogous to the realistic proportion of the implant [6]. The models were checked as volumetric solids containing a standard prosthetic platform with a morse-taper connection (Fig 1).

The dental crown was simulated based on a previously designed volumetric model of a complete molar [14]. A high resolution micro-CT scanner system (Bruker micro CT) was used to produce the 3D shell of the lower molar [6, 14]. With InVesalius 3.1.1 software, the data groups were processed and polysurfaces were created with cross-section polylines. The parametric size was then determined by means of loft-connected polysurfaces. The crown´s dimensions, after post-processing the model, were 12.3 mm (mesio-distally) and 10.6 mm (bucco-lingually).

Following a previous investigation, a basic jawbone structure was selected (Fig 1). For that, the bone model was reduced and individualized into a cylinder form (15 mm x 20 mm). Based on the bone density properties of the literature, a D2 bone type was designed containing 2.0 mm cortical thickness juxtaposed with the trabecular bone tissue. To guarantee a correct connection at bone implant contact (BIC), a Boolean difference was performed, by the difference between the implant and bone volume [6, 17]. Based on that, an ideal condition was assumed with total osseointegration of the implant.

The final geometries were imported into computer-aided engineering software (ANSYS 19.2, ANSYS Inc., Houston, TX, USA) in STEP format. The meshing process was created using tetrahedral elements (Fig 2), after the subsequent iterative mesh refinement procedure of convergence [10].

Elastic modulus and Poisson ratios for each component were assigned to each structure, considering linear, elastic, homogeneous and isotropic behavior (Table 1).

The model wax fixed at the bone surface and a load of 600 N was applied to simulate the occlusal force at the upper surface of the food bolus created upon the 3D coordinate system (Fig 3).

## Results

According to the evaluated factors, von-Mises Stress (MPA) maps were calculated to evaluate each situation. The section plane for the crowns showed a similar stress pattern among the models, apparently without a qualitative difference between them when containing or not the prosthetic screw (Fig 4). Each stress map was based on a color-coded nonlinear scale of stress ranging from -13 until 52 MPa for the crown, 8–117 MPa for the abutment, 0–91 MPa for the prosthetic screw, and 7–123 MPa for the implant fixture. Yet, it is possible to notice a difference in the stress concentrated at the intaglio surface. In the simulated scenario, the lower the stiffness of the crown, the lower the stress peak inside of it.

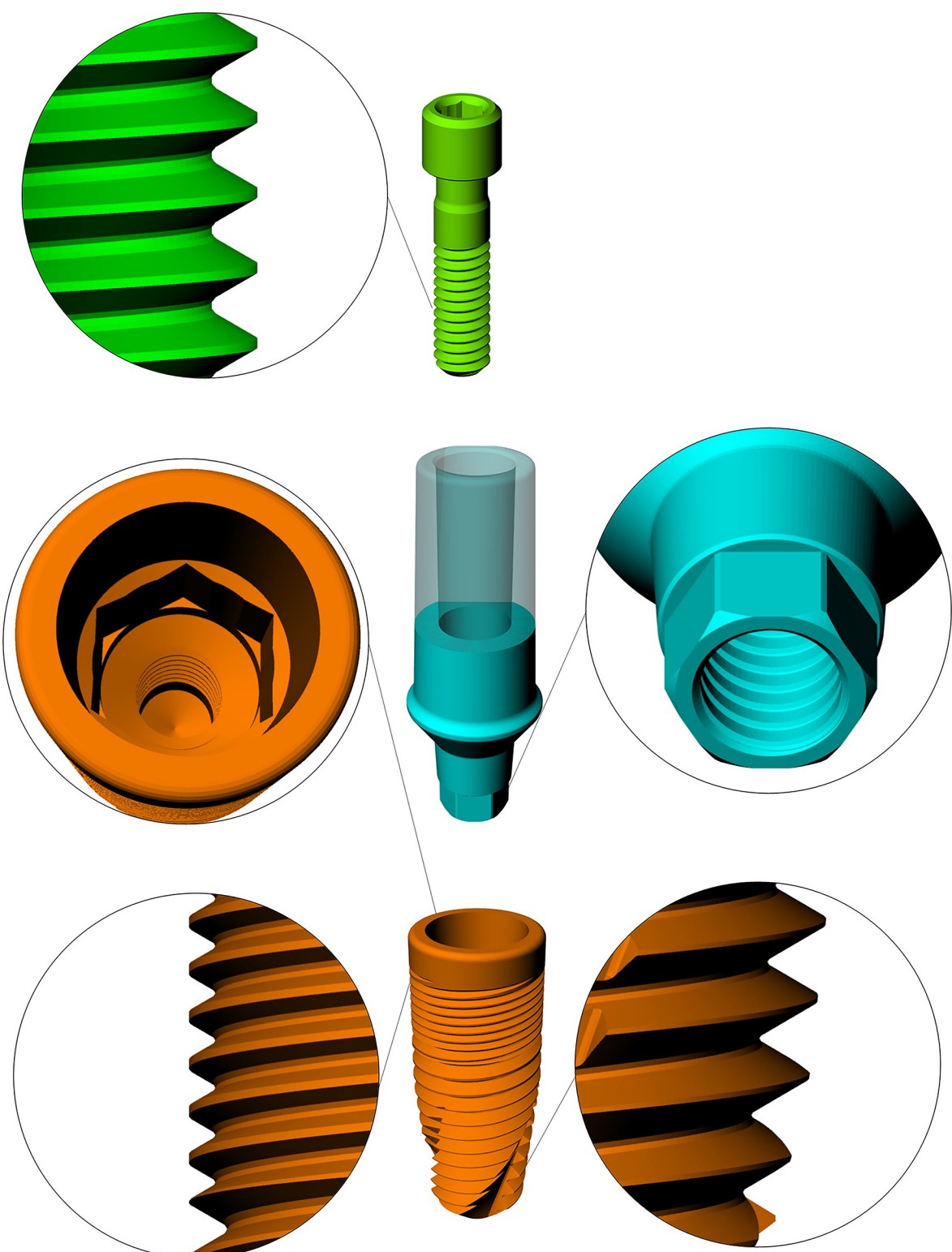

**Fig 1. Three-dimensional files from the manufacturer showing the different fixture features in the CAD software.** In this model, the height of the abutment was sectioned according to the size of the crown according to the manufacturer's recommendation.

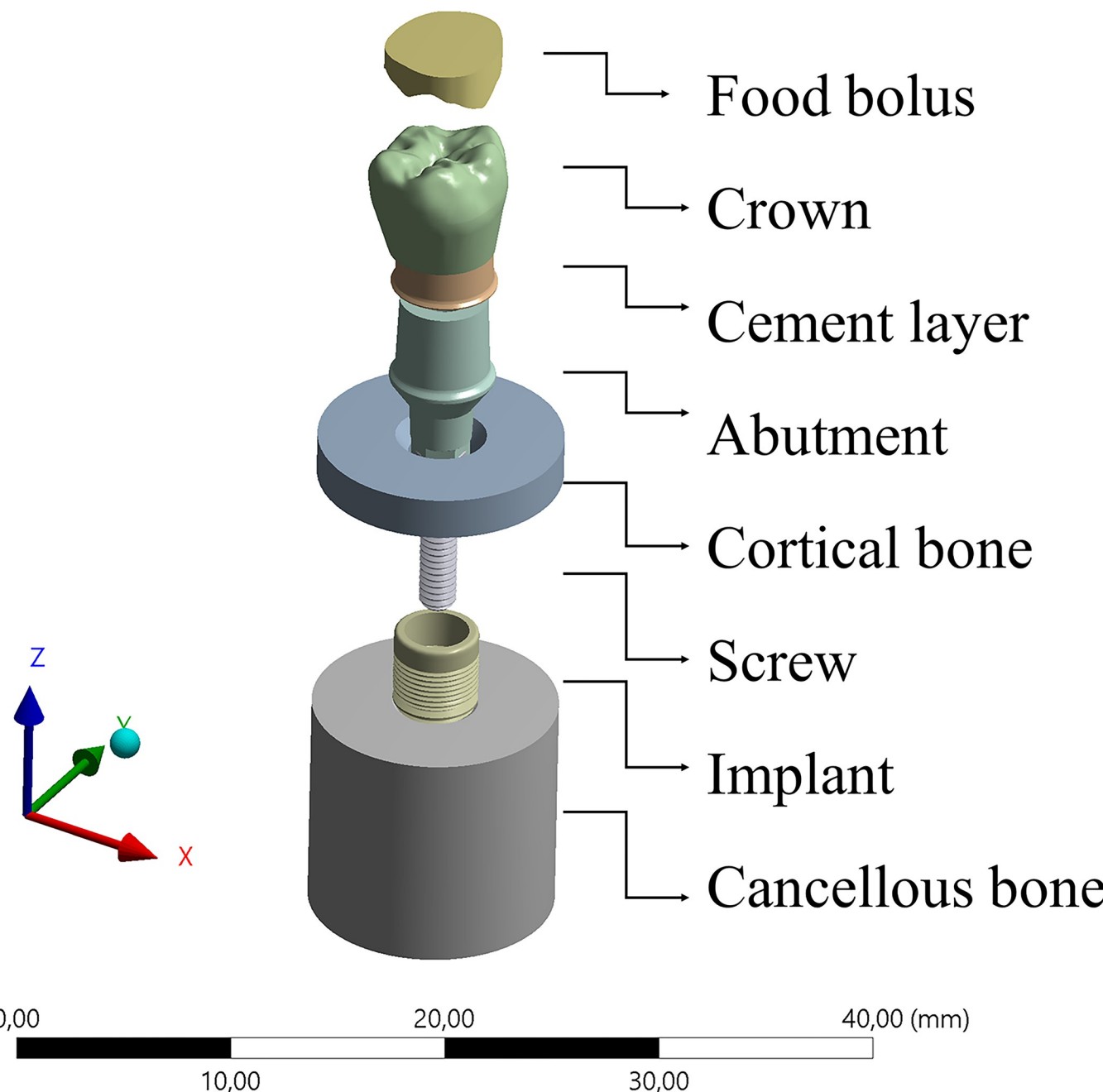

Food bolus

Crown

Cement layer

Abutment

Cortical bone

Screw

Implant

Cancellous bone

**Fig 2. Numerical model after NURBS modelling containing different volumetric structures.**

Additionally, a further dissimilar mechanical response was noticed for the stress concentration trend between the abutment, with the highest stress magnitude calculated at the region of cervical level and occlusal surface (Fig 5). While flexible crowns concentrated less stress on its structure, they deformed largely, stressing the abutment more than zirconia crowns. The effect of prosthetic screw is visible at the connection region, reducing the stress when present inside of the abutment.

When comparing both prosthetic screws between block-composite and zirconia crowns, there are evident differences in mechanical response. For both models, the region of highest stress magnitude was the screw neck and the first threads (Fig 6).

**Table 1. Mechanical properties of the materials simulated in this study.**

| Material | Elastic modulus (GPa) | Poisson ratio |
| --- | --- | --- |
| Titanium | 110 | 0.3 |
| Zirconia | 200 | 0.3 |
| Estelite P Block (Block composite resin) | 13.8 | 0.3 |
| Resin cement | 5 | 0.3 |
| Cortical bone | 13.7 | 0.3 |
| Trabecular bone | 5.5 | 0.3 |

The implants showed a different mechanical behaviour when considering the presence or absence of the prosthetic screw, but not when considering different crown materials (Fig 7). Like the abutment structure, when the screw was present, less stress was concentrated on the cervical side of the implant, and more stress is detectable in the threaded region.

The microstrain maps in the peri-implant tissue is visible in Fig 8. The micro-strain peaks are reported in Table 2 together with the stress peaks per region. Regardless the simulated condition, all simulated models were capable to dissipate the load at the bone-implant interface with a comparable pattern, showing that nether crown material or prosthetic screw presence would affect the bone mechanical behavior. Taking 3000 με as the standard strain for bone resorption, it was possible to assent that, in any case, the calculated strains did not promote any effect of bone resorption.

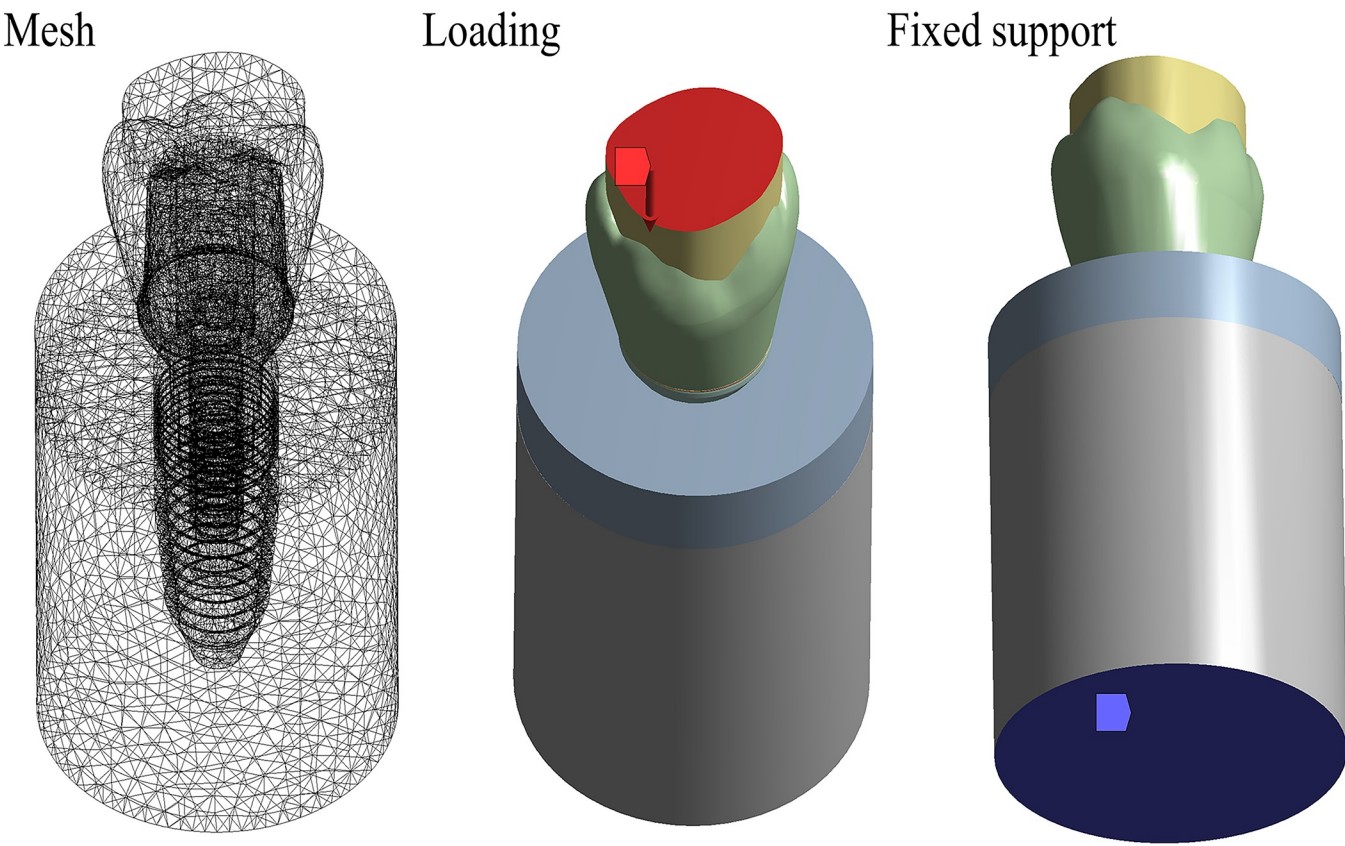

**Fig 3. Meshing process and boundary conditions simulated in the present study.**

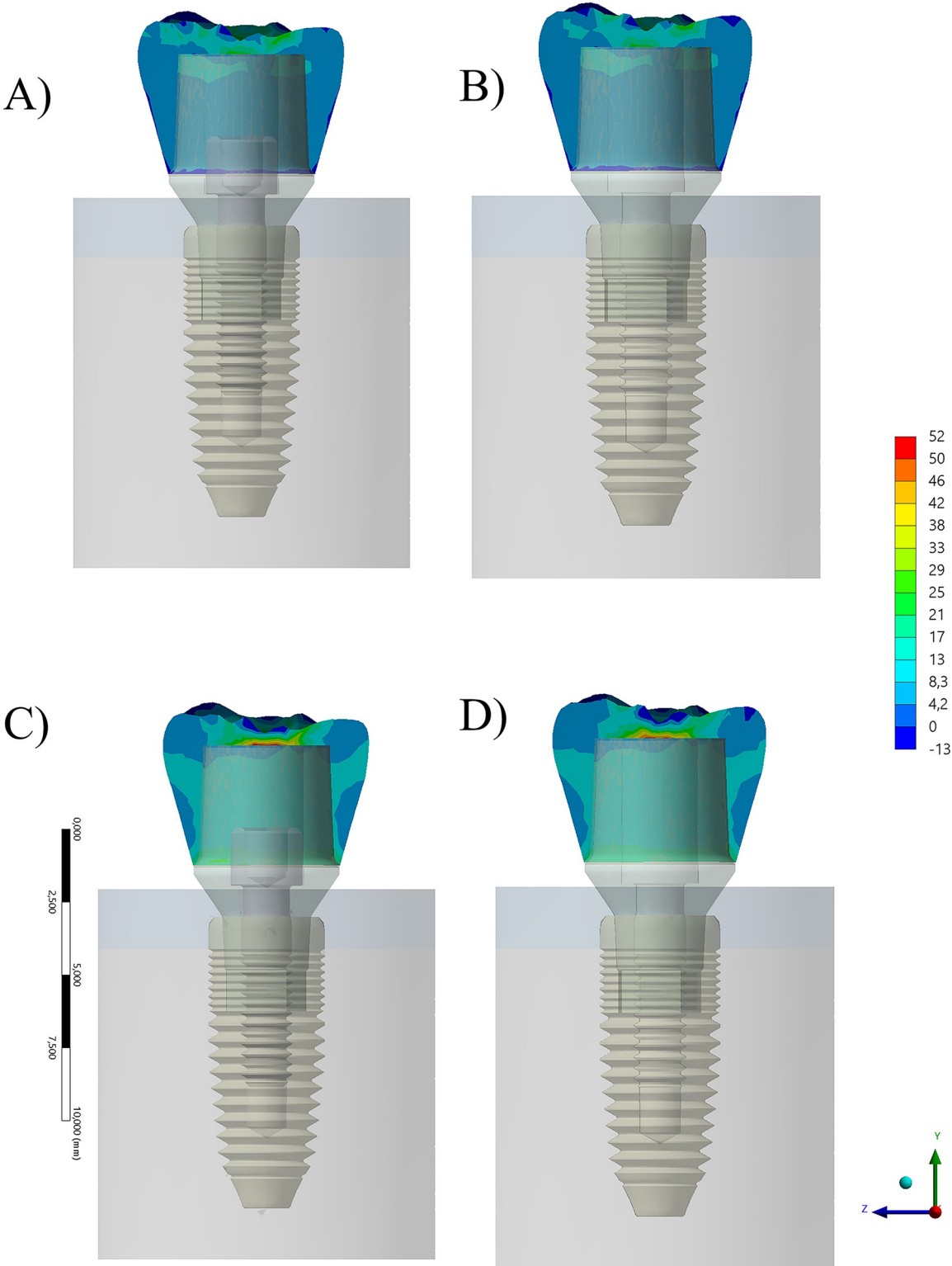

**Fig 4. Section plane for von-Mises Stress contour plots for the crown according to different conditions.** A) Composite resin with prosthetic screw, B) Composite resin without prosthetic screw and C) Zirconia with prosthetic screw and D) Zirconia without prosthetic screw.

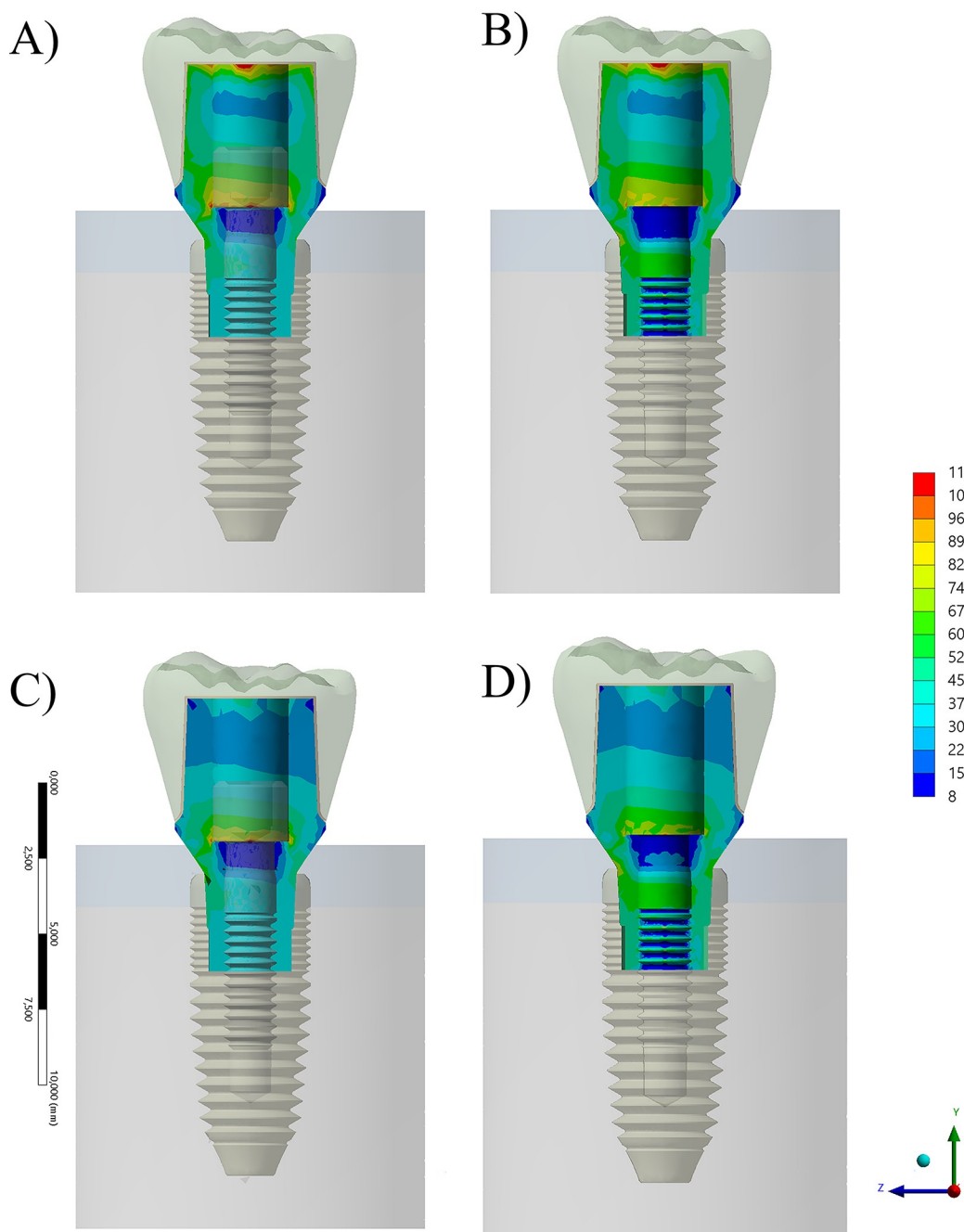

**Fig 5. Section plane for von-Mises Stress contour plots for the abutment according to different conditions.** A) Composite resin with prosthetic screw, B) Composite resin without prosthetic screw and C) Zirconia with prosthetic screw and D) Zirconia without prosthetic screw.

## Discussion

Morse-taper implants are a design of dental implant system that uses a tapered connection between the implant and the abutment. The Morse-taper connection was originally developed in the 19th century for use in machine tools and has since been adapted for use in dental implants [20, 21]. Morse-taper implants offer several advantages over other implant systems:

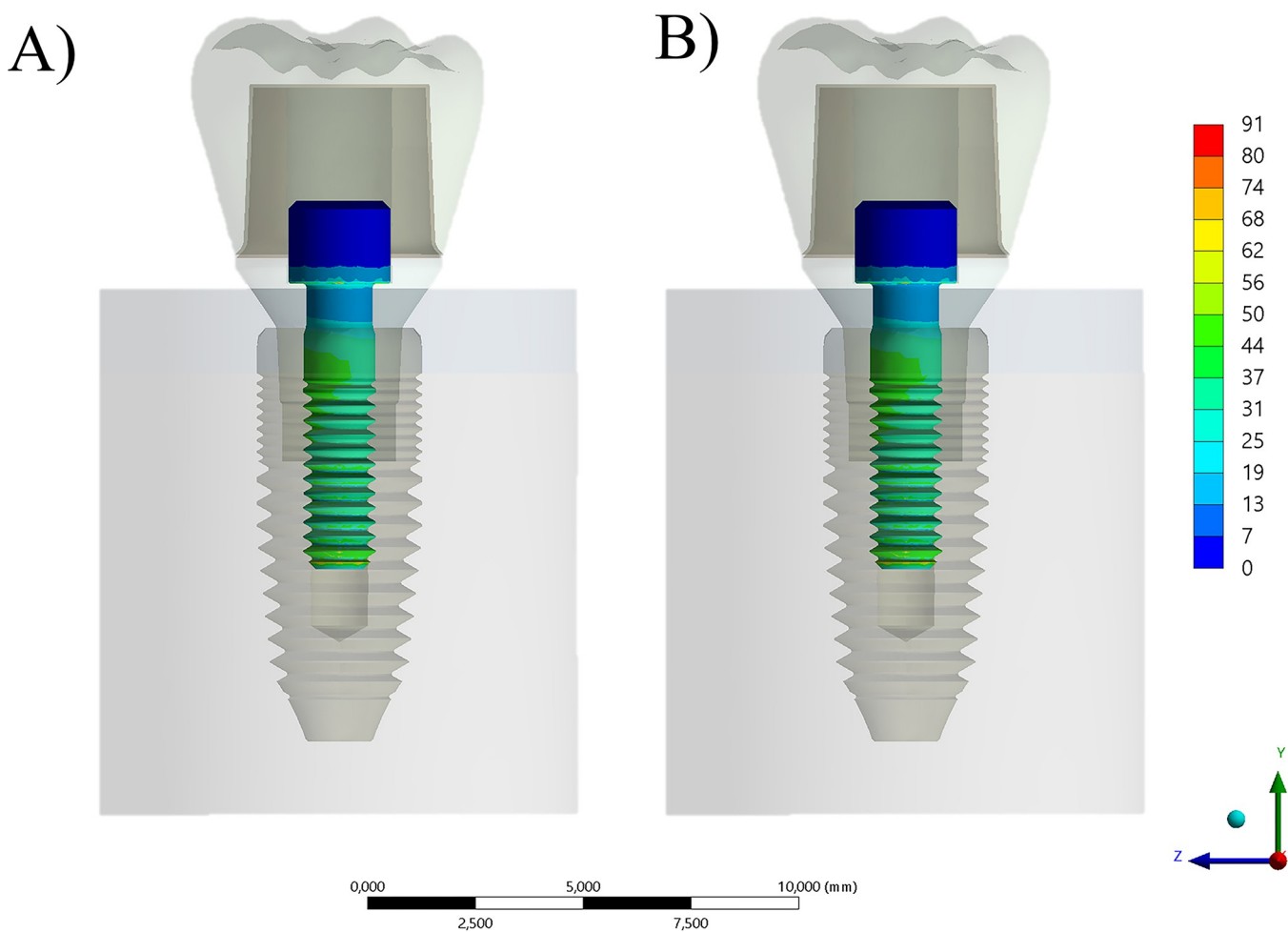

**Fig 6. Section plane for von-Mises Stress contour plots for the prosthetic screw (in the models that contain this structure) according to different conditions.** A) Composite resin with prosthetic screw, B) zirconia with prosthetic screw.

they provide a strong and stable interface between the implant and the abutment, which can increase the longevity of the restoration [19–22]. Additionally, Morse-taper implants can be placed using a single-stage surgical procedure, which can reduce overall treatment time and improve patient comfort [19–23].

Prosthetic screw failure can occur in dental implant treatment when the screw that connects the dental prosthesis to the implant becomes loose or fractured [24]. This can lead to implant instability, implant fracture, or other complications that can compromise implant treatment success. To limit fixation screw complications, an alternative has been manufactured to screw-retained implant systems (ie, implant abutment connections without screws) [23]. There are several potential causes of prosthetic screw failure, including inadequate tightening torque, incorrect screw positioning, misalignment of implant components, fatigue or corrosion of screws, or excessive forces on the prosthesis [24]. The present study complements this information, showing that the abutment screw concentrate stress during loading. So, the both null hypotheses have been rejected: the screw effect inside the connection is negligible; 2) crown's material modulus does not influence the implant-bone interface.

Solid abutments are a type of dental implant abutment that is made from a single piece of material, designed to provide a stable, strong foundation for dental restorations, such as

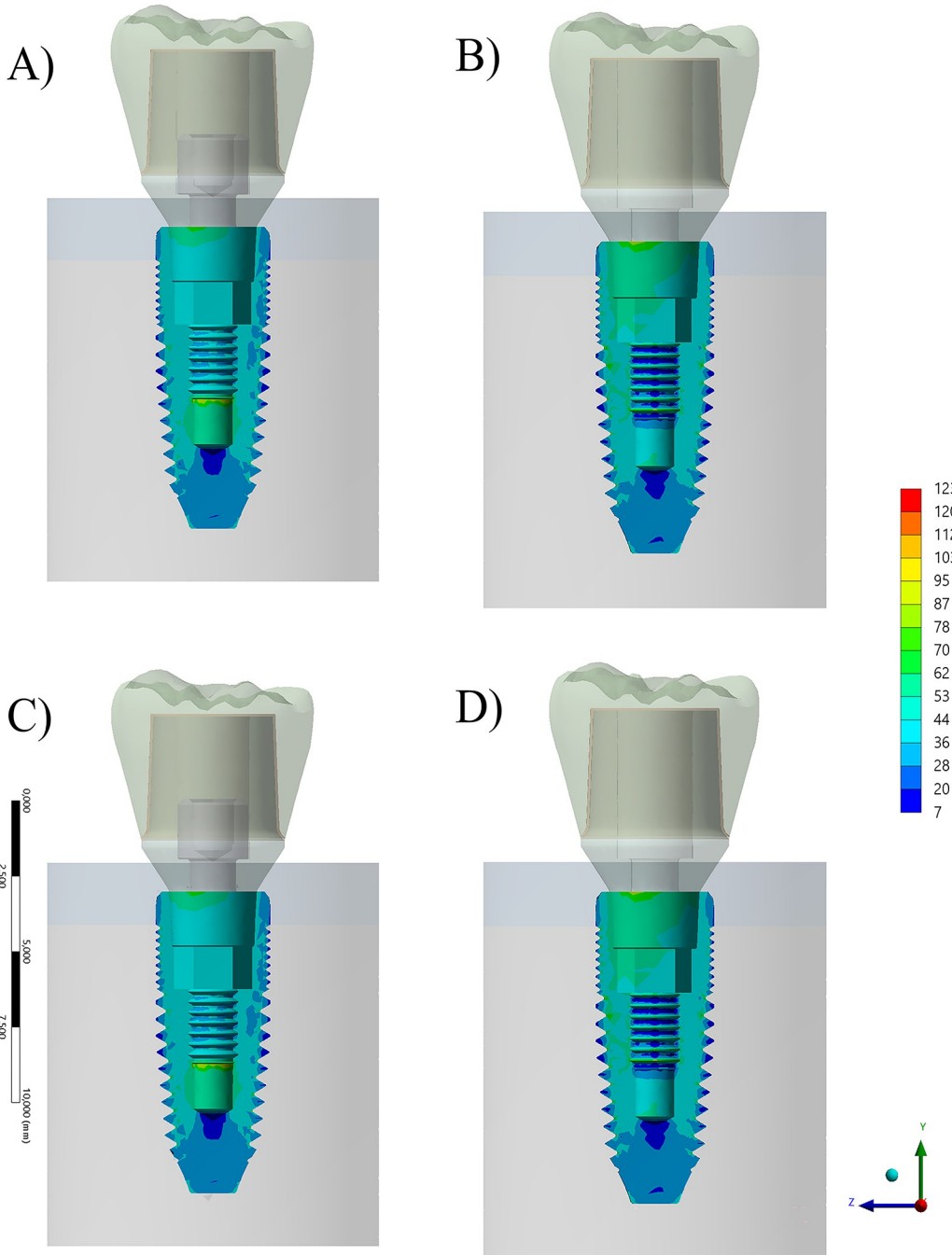

**Fig 7. Section plane for von-Mises Stress contour plots for the implant according to different conditions.** A) Composite resin with prosthetic screw, B) Composite resin without prosthetic screw and C) Zirconia with prosthetic screw and D) Zirconia without prosthetic screw.

crowns or bridges that are attached to dental implants [25]. Because they are made from a single piece of material, they are less prone to mechanical failure or loosening than multi-piece abutments. However, solid abutments may not be appropriate for all patients or implant systems. In addition, solid abutments can still have threads in their structure being more difficult to remove than other types of abutments, making them less suitable for patients who may need future implant follow-up [25, 26].

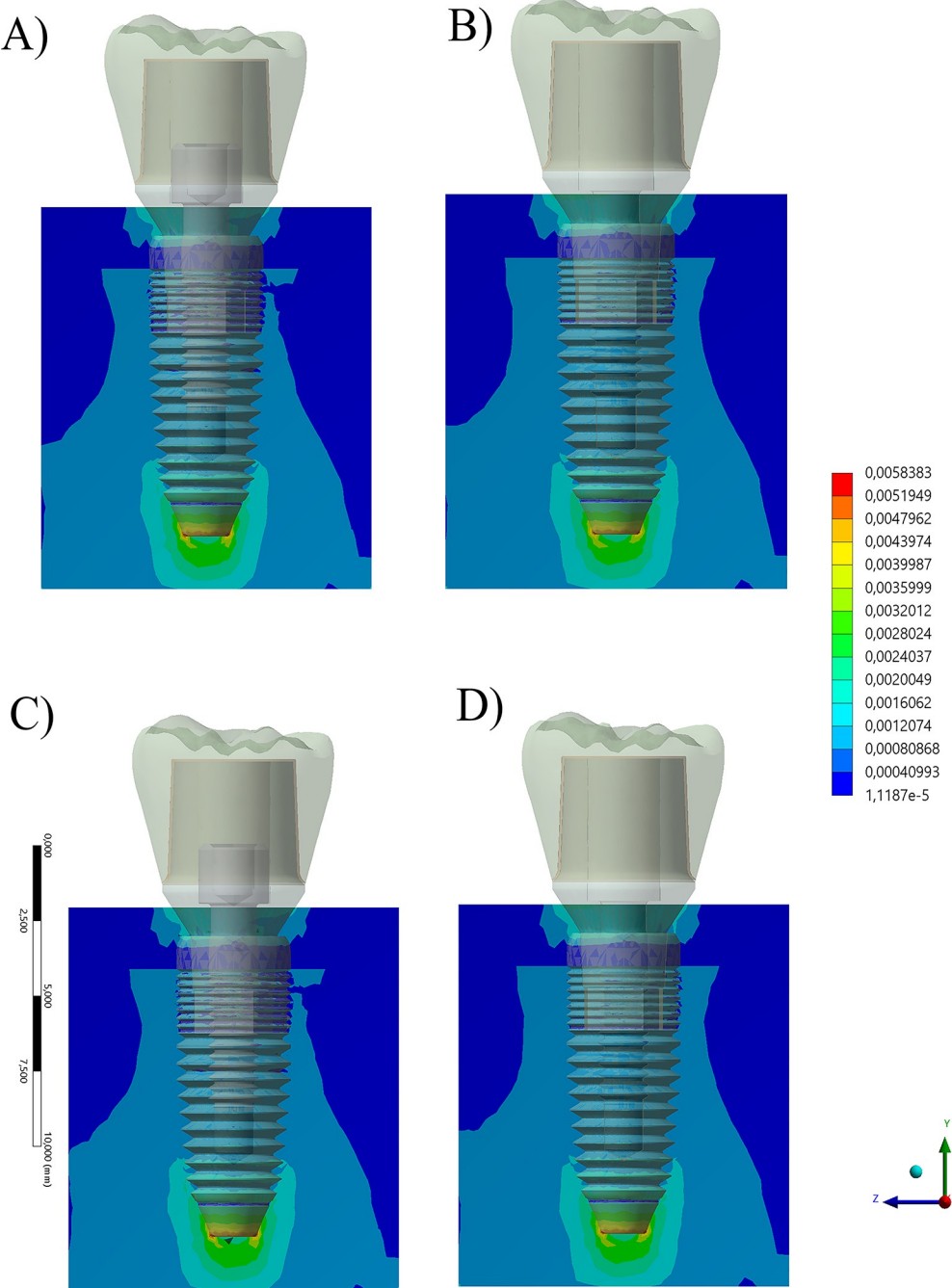

**Fig 8. Section plane for equivalent strain contour plots for the bone tissue according to different conditions.** A) Composite resin with prosthetic screw, B) Composite resin without prosthetic screw and C) Zirconia with prosthetic screw and D) Zirconia without prosthetic screw.

In this sense, abutments without screws can theoretically offer several advantages over traditional screw-retained abutments. Because there is no screw, the abutment can be designed with a more anatomically contoured shape, which can improve the aesthetics and function of the implant-supported prosthesis [26]. Furthermore, non-screw abutments can eliminate the risk of screw loosening or component failure, which can improve the long-term stability and

**Table 2. Stress peaks per region and microstrain in bone tissues for each evaluated model.**

| Crown material | Prosthetic screw | Crown | Abutment | Implant | Screw | Bone tissue |
|---|---|---|---|---|---|---|
| Zirconia | Yes | 64.47 MPa | 99.71 MPa | 112.74 MPa | 88.74 MPa | 8.3023e-004 mm/mm |
| Zirconia | No | 64.47 MPa | 126.63 MPa | 124.37 MPa | - | 8.6622e-004 mm/mm |
| Composite resin | Yes | 52.20 MPa | 92.28 MPa | 112.6 MPa | 90.87 MPa | 8.2918e-004 mm/mm |
| Composite resin | No | 52.21 MPa | 114.25 MPa | 124.18 MPa | - | 8.6482e-004 mm/mm |

success of implant treatment [23]. In the present study, the bone tissue was similar between models with and without prosthetic screw, regardless the crown's material.

Based on the reported information, a study purpose that the conical abutment alone to secure the implant–abutment connection can eliminate the possibility of screw loosening and fracture [23]. According to the reported in-vitro study, the survival rates of screw-retained and screwless abutments are similar. The authors informed that the use of a screwless morse-taper implant–abutment connection represents a valid form of treatment for single-tooth replacement. According to their experiment, the used implants withstand out the average occlusal forces even after an extended interval of artificial loading [23]. However, this is not a consensus in the literature; another study found that the mechanical resistance of the screwless morse-taper implant system is lower than that of the internal screw-retained implant systems, which could result in more frequent clinical complications [22]. The present study showed that the presence of a prosthetic screw is significant in reducing stress in the abutment connection but increases stress in the implant. There was no difference for the crown and bone tissue.

Another parameter evaluated in this study was the crown material. When an implant is placed in the bone, it can be subject to a variety of forces, including chewing and bite forces, as well as other stresses caused by the oral environment [2]. The crown that is placed on top of the implant must be able to withstand these forces without causing damage to the implant or the surrounding tissue [3–5]. The elastic modulus of a dental crown refers to its ability to deform under stress, and it can play an important role in the success of implant therapy [4, 11, 27]. In general, materials with a higher elastic modulus, such as zirconia, are expected to be less likely to transfer stress to the implant and surrounding tissue, which can help reduce the risk of implant failure or complications [4, 11, 27]. However, other studies showed that due to the presence of cement layer, abutment, screw and other components from the implant-supported restoration, the crown's effect at the bone level is usually insignificant [27–29]. The present study corroborates with them, showing a similar stress pattern between both materials.

While zirconia is often chosen for its durability, the choice of restorative material should be based on several factors, including the individual patient's needs and circumstances, the clinical requirements of the implant site, and the occlusion and bite forces of the patient [30–33]. Block composite crowns offer several advantages for implant-supported restorations: they are less expensive, require fewer post-processing steps, and can be customized to match the color and shape of the patient's natural teeth, which helps to create a natural-looking smile; they are also relatively easy to repair or replace if they become damaged or worn over time [31, 32].

One potential disadvantage of block-composite crowns for implant-supported restorations is that they may not be as durable as other materials such zirconia [33–35]. They may also be more prone to chipping or cracking if exposed to excessive biting forces or if the patient grinds their teeth. Overall, composite CAD/CAM crowns can be a good option for implant-supported restorations in certain cases, particularly for anterior teeth or when the antagonist is a composite-restored tooth [34, 35].

The biomechanics of implant-supported restorations is an important consideration in the design and placement of dental implants to ensure long-term success and stability of the

restoration [36]. The biomechanics of implant-supported restorations involve interactions between the dental implant that serves as an artificial tooth root, the surrounding bone, and the artificial tooth or denture [4]. The distribution of forces is important because excessive stress can lead to bone resorption or implant failure over time [5]. Several factors influence the biomechanics of implant-supported restorations, including the location of the implant in the bone, the number and distribution of implants used, the shape and size of the implant, the type of attachment used to connect the implant to the artificial tooth or denture, and the occlusal forces generated during chewing [4–7, 10, 17–23, 25–27, 37, 38]. Additionally, the present results showed that the presence of screw and crown stiffness also can affect the implant mechanical behavior, however the first is more significant than the second factor. In this sense the null hypotheses were rejected.

This study has certain limitations that need to be considered. Firstly, the force applied in the simulation was unidirectional, whereas forces from other regions may generate different outcomes. Moreover, the elastic modulus was isotropic, which is not the case with human tissue [38, 39]. Also, there was no consideration of external factors such as saliva, pH variation, temperature variation, or the presence of different antagonist materials. Future studies should investigate these factors to understand the mechanical effect on the implant-supported crown. Furthermore, the materials were considered ideals, without defects on their structure as well as with ideal contacting surfaces. Despite these limitations, the study provides a numerically controlled experiment that shows proportionality stress states that can be compared quantitatively and qualitatively. However, further investigations are required to corroborate or not with the present theoretical findings.

## Conclusions

Based on this linear analysis, within the limits of this investigation, it is suggested that the absence of a prosthetic screw increases stress inside the abutment and implant models, without effect on the crown and bone tissue. Stiffer crowns concentrate more stress on their structure, reducing the amount of stress on the abutment.

## Author Contributions

**Conceptualization:** Pietro Ausiello, João Paulo Mendes Tribst.

**Data curation:** Pietro Ausiello, João Paulo Mendes Tribst.

**Investigation:** Ettore Epifania, Alessia Mancone.

**Methodology:** Ettore Epifania, Pietro Ausiello.

**Project administration:** Alessandro E. di Lauro.

**Resources:** Alessandro E. di Lauro.

**Supervision:** Franklin Garcia-Godoy.

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
