## [Decision Letter · Decision Letter 0]

4 Apr 2023

PONE-D-23-06865

Effect of crown stiffness and prosthetic screw absence on the stress distribution in implant-supported restoration: A 3D finite element analysis.

PLOS ONE

Dear Dr. ausiello,

Thank you for submitting your manuscript to PLOS ONE. After careful consideration, we feel that it has merit but does not fully meet PLOS ONE’s publication criteria as it currently stands. Therefore, we invite you to submit a revised version of the manuscript that addresses the points raised during the review process.

We look forward to receiving your revised manuscript.

Kind regards,

Artak Heboyan, Ph.D.

Academic Editor

PLOS ONE

Journal Requirements:

5. Please remove your figures from within your manuscript file, leaving only the individual TIFF/EPS image files, uploaded separately. These will be automatically included in the reviewers’ PDF.

6. We note you have included a table to which you do not refer in the text of your manuscript. Please ensure that you refer to Table 2 in your text; if accepted, production will need this reference to link the reader to the Table.

7. Please include a copy of Table 3 which you refer to in your text on page 14.

Additional Editor Comments:

Dear authors,

Thank you for submitting your paper to PLOS ONE.

After careful evaluation of your paper, we found that it can be accepted for publication after appropriate revision. Please, find attached the reviewers` comments and address all their concerns very carefully. After revision we will evaluate it for further processing.

Regards

Reviewers' comments:

Reviewer's Responses to Questions

**Comments to the Author**

1. Is the manuscript technically sound, and do the data support the conclusions?

Reviewer #1: Yes

Reviewer #2: Yes

Reviewer #3: Yes

2. Has the statistical analysis been performed appropriately and rigorously? 

Reviewer #1: No

Reviewer #2: Yes

Reviewer #3: N/A

3. Have the authors made all data underlying the findings in their manuscript fully available?

Reviewer #1: No

Reviewer #2: Yes

Reviewer #3: Yes

4. Is the manuscript presented in an intelligible fashion and written in standard English?

Reviewer #1: Yes

Reviewer #2: Yes

Reviewer #3: Yes

5. Review Comments to the Author

Reviewer #1: Dear authors,

it was evaluated the article “Effect of crown stiffness and prosthetic screw absence on the stress distribution in implant-supported restoration: A 3D finite element analysis”.

Aim: “considered the effects of occlusal loading on strain and stress development in a Morse-type implant-abutment connection where taper surfaces are connected by a cold-welding effect and the two parts engaged together, dependently on different crown material stiffness (E)”.

“The null hypothesis were: 1) screw effect inside the connection is negligible; 2) crown restoring material modulus doesn’t influence the implant-bone interface.”

It is an interesting article, although I considered it simple and straightforward. Some concerns and comments were done (below).

———

INTRO

- lines 85-88: The paragraph can be better developed. I recommend to include also a recent article: Three-Dimensional FEA Analysis of the Stress Distribution on Titanium and Graphene Frameworks Supported by 3 or 6-Implant Models. Biomimetics 2023, 8, 15. https://doi.org/10.3390/biomi- metics8010015

- line 92: avoid contraction: “doesn’t”

M&M

- please, include the statistical analysis (subtopic) to show the significance of the study. Without this subtopic is not possible to publish the article.

Results

- To include the statistically significant results for the data presented

Discussion - ok!

Conclusion - responded the hypothesis done. Therefore, without statistics is not possible to say there was significant the results.

- line 290: “within the limits of this investigation” - where were shown the limitations?

Reviewer #2: The study has been conducted in a proper way, but needs slight corrections as mentioned below:

1. The keywords should be written in a alphabetical order and as per MeSH guidelines.

2. There are few spelling mistakes, like Implant etc..

3. The discussion needs to be elaborated with comparison of the results of the present study with the studies earlier conducted on the similar lines.

4. The references should include some recently conducted studies on the similar lines.

5. Rest everything seems to be fine.

Reviewer #3: Dear Authors,

I read the manuscript with great interest. The manuscript under review attempts to evaluate the effect of crown stiffness and prosthetic screw absence on the stress distribution in implant-supported restoration using A 3D finite element analysis. In general, the manuscript captures details of the study design and implementation of the project. All the sections of the manuscript are well written and concluded. The study is of sound design and of clear practical and clinical interest. Kindly find below the detailed comments

Abstract:

1. Kindly provide the statistical significant values

2. Keywords : correct the spelling mistake (Dental Implant) and kindly provide the MesH keywords

Methods:

1. Kindly shift Fig 2 before the table 1, as its cited first

2. Kindly provide a conceptual framework for methods or illustrated to make reader easy

3. Provide the calibrations

4. Table I : kindly mention the complete form and describe the unit GPa in footer or table legend

Results:

1. Kindly provide the range and unit for color-coded non-linear scale of stress

2. For all figures

3. Table 2 : kindly mention the complete form and describe the unit MPa in footer or table legend

Discussion:

1. Kindly write limitations of the current study

2. Kindly write briefly on Clinical implementation, future recommendations

Best regards

6. PLOS authors have the option to publish the peer review history of their article (what does this mean?). If published, this will include your full peer review and any attached files.

Reviewer #1: **Yes: **Gustavo Fernandes

Reviewer #2: No

Reviewer #3: No

---

## [Author Response · Author response to Decision Letter 0]

10 Apr 2023

Additional Editor Comments:

Dear authors,

Thank you for submitting your paper to PLOS ONE.

After careful evaluation of your paper, we found that it can be accepted for publication after appropriate revision. Please, find attached the reviewers` comments and address all their concerns very carefully. After revision we will evaluate it for further processing.

Regards

Dear editor, thank you for considering the present study for possible publication in PLOS One. We would like to express our sincere gratitude for all thoughtful and constructive critiques, which has enabled us to refine our ideas and strengthen our arguments. The reviewers’ expertise in the field has been invaluable to us, and we are grateful for your time and effort in providing us with such thorough and insightful feedback.

Comments to the Author

Reviewer #1: Dear authors,

it was evaluated the article “Effect of crown stiffness and prosthetic screw absence on the stress distribution in implant-supported restoration: A 3D finite element analysis”.

Aim: “considered the effects of occlusal loading on strain and stress development in a Morse-type implant-abutment connection where taper surfaces are connected by a cold-welding effect and the two parts engaged together, dependently on different crown material stiffness (E)”.

“The null hypothesis were: 1) screw effect inside the connection is negligible; 2) crown restoring material modulus doesn’t influence the implant-bone interface.”

It is an interesting article, although I considered it simple and straightforward. Some concerns and comments were done (below).

Dear reviewer, we appreciate the time and effort to evaluate this study. I hope that we had answered all the points and improved the manuscript as expected.

———

INTRO

- lines 85-88: The paragraph can be better developed. I recommend to include also a recent article: Three-Dimensional FEA Analysis of the Stress Distribution on Titanium and Graphene Frameworks Supported by 3 or 6-Implant Models. Biomimetics 2023, 8, 15. https://doi.org/10.3390/biomi- metics8010015

- line 92: avoid contraction: “doesn’t”

The text has been reviewed and the English language checked. Relevant references have been included in the manuscript, including the suggested one.

M&M

- please, include the statistical analysis (subtopic) to show the significance of the study. Without this subtopic is not possible to publish the article.

Results

- To include the statistically significant results for the data presented

Dear reviewer, we appreciate your suggestion. However, for finite element studies the difference between models is assumed as definitive and the statistical approach is not necessary. It happens, because FEA is already a numerical model and using other numerical model to show difference between it would not be indicate. You can check other studies in this field with similar method to see that mostly of them, correctly, does not apply statics.

Discussion - ok!

Conclusion - responded the hypothesis done. Therefore, without statistics is not possible to say there was significant the results.

- line 290: “within the limits of this investigation” - where were shown the limitations?

A new paragraph has been added describing the major limitations.

Reviewer #2: The study has been conducted in a proper way, but needs slight corrections as mentioned below:

1. The keywords should be written in a alphabetical order and as per MeSH guidelines.

They have been corrected in the system.

2. There are few spelling mistakes, like Implant etc..

The English language has been corrected in this new version.

3. The discussion needs to be elaborated with comparison of the results of the present study with the studies earlier conducted on the similar lines.

New references have been added and the text improved.

4. The references should include some recently conducted studies on the similar lines.

New references have been added and the text improved.

5. Rest everything seems to be fine.

Thank you for your appreciated comments.

Reviewer #3: Dear Authors,

I read the manuscript with great interest. The manuscript under review attempts to evaluate the effect of crown stiffness and prosthetic screw absence on the stress distribution in implant-supported restoration using A 3D finite element analysis. In general, the manuscript captures details of the study design and implementation of the project. All the sections of the manuscript are well written and concluded. The study is of sound design and of clear practical and clinical interest. Kindly find below the detailed comments.

Thank you for taking the time to review our manuscript. We appreciate your valuable feedback and comments, which have helped to improve the quality of our work. Your insights and suggestions have been instrumental in shaping our manuscript.

Abstract:

1. Kindly provide the statistical significant values

Finite element analysis (FEA) is a computational tool used to analyze the behavior of complex engineering systems under different conditions. FEA results are obtained by solving a set of equations using mathematical models and numerical methods. These results are usually expressed in terms of stresses, strains, displacements, and other parameters of interest.

FEA results are not based on samples or populations, unlike statistical analyses. Therefore, traditional statistical tests such as t-tests, ANOVA, or regression analysis are not applicable to evaluate FEA results.

Instead, FEA results are typically evaluated based on engineering principles, such as stress and strain levels relative to the material's yield strength or the deformation limits of the component being analyzed. These techniques can help to identify the most influential factors in the model and their relative importance on the output parameters of interest.

2. Keywords : correct the spelling mistake (Dental Implant) and kindly provide the MesH keywords

Keywords have been corrected in the system.

Methods:

1. Kindly shift Fig 2 before the table 1, as its cited first

The correction has been performed as suggested.

2. Kindly provide a conceptual framework for methods or illustrated to make reader easy

3. Provide the calibrations

We would appreciate if you be more specific in topics 2 and 3.

4. Table I : kindly mention the complete form and describe the unit GPa in footer or table legend

Table 1 contains the unit GPa in the column “Elastic modulus”.

Results:

1. Kindly provide the range and unit for color-coded non-linear scale of stress

2. For all figures

This information has been added to the text.

3. Table 2 : kindly mention the complete form and describe the unit MPa in footer or table legend

Units are present in table 2.

Discussion:

1. Kindly write limitations of the current study

A new paragraph has been added with this information

2. Kindly write briefly on Clinical implementation, future recommendations

A new paragraph has been added with this information

Best regards

---

## [Editor Report · Decision Letter 1]

24 Apr 2023

Effect of crown stiffness and prosthetic screw absence on the stress distribution in implant-supported restoration: A 3D finite element analysis.

PONE-D-23-06865R1

Dear Dr. ausiello,

We’re pleased to inform you that your manuscript has been judged scientifically suitable for publication and will be formally accepted for publication once it meets all outstanding technical requirements.

Kind regards,

Artak Heboyan, Ph.D.

Academic Editor

PLOS ONE
---

## [Editor Report · Acceptance letter]

26 Apr 2023

PONE-D-23-06865R1 

Effect of crown stiffness and prosthetic screw absence on the stress distribution in implant-supported restoration: A 3D finite element analysis. 

Dear Dr. ausiello:

I'm pleased to inform you that your manuscript has been deemed suitable for publication in PLOS ONE. Congratulations! Your manuscript is now with our production department. 

Kind regards, 

on behalf of

Dr. Artak Heboyan 

Academic Editor

PLOS ONE